# A Boundary Forcing Sensitivity Analysis of the West African Monsoon Simulated by the Modèle Atmosphérique Régional

**Guillaume Chagnaud \*, Hubert Gallée, Thierry Lebel, Gérémy Panthou and Théo Vischel**

Université Grenoble Alpes, IRD, CNRS, Grenoble INP, IGE, 38000 Grenoble, France;
hubert.gallee@univ-grenoble-alpes.fr (H.G.); thierry.lebel@univ-grenoble-alpes.fr (T.L.);
geremy.panthou@univ-grenoble-alpes.fr (G.P.); theo.vischel@univ-grenoble-alpes.fr (T.V.)

\* Correspondence: guillaume.chagnaud@univ-grenoble-alpes.fr

**Abstract:** The rainfall regime of West Africa is highly variable over a large range of space and time scales. With rainfall agriculture being predominent in the region, the local population is extremely vulnerable to intraseasonal dry spells and multi-year droughts as well as to intense rainfall over small time steps. Were this variability to increase, it might render the area close from becoming unhabitable. Anticipating any change is thus crucial from both a societal and a scientific perspective. Despite continuous efforts in Global Climate Model (GCM) development, there is still no agreement on the sign of the future rainfall regime change in the region. Regional Climate Models (RCMs) are used for more accurate projections of future changes as well as end-user-oriented impact studies. In this study, the sensitivity of the Modèle Atmosphérique Régional (MAR) to homogeneous perturbations in boundary forcing air temperature and/or SST is assessed with the aim to better understand (i) the thermodynamical imprint of the recent rainfall regime changes and (ii) the impact of errors in driving data on the West African rainfall regime simulated by an RCM. After an evaluation step where the model is proved to satisfactorily simulate the West African Monsoon (WAM), sensitivity experiments display contrasted, sizable and robust responses of the simulated rainfall regime. The rainfall responses to the boundary forcing perturbations compare in magnitude with the intrinsic model bias, giving support for such an analysis. A physical interpretation of the rainfall anomalies provides confidence in the model response consistency and shows the potential of such an experimental protocol for future climate change downscaling over this region.

**Keywords:** West African Monsoon; precipitation; regional climate model; sensitivity analysis

## 1. Introduction

### 1.1. General Context

Monsoon climate systems are key components of the Earth climate system that are mildly reproduced by global climate models, especially regarding rainfall [1]. Yet, they control the hydrology of very populated regions, where both water resources scarcity and water-related risks—such as flash floods or large scale floodings—are a permanent threat to life. The West African Monsoon (WAM) is emblematic of these dual scientific and socio-economic challenges. It is characterized by strong fluctuations of its rainfall regime, from multi-decadal to sub-daily time scales, with an immediate hydrological effect (see e.g., [2,3]) as well as numerous socio-economic consequences regarding water resources, food security and health. The continuous drought that struck the region from the end of the 1960s to the end of the 1990s has shown how vulnerable populations are to a lasting rainfall deficit, generating starvations and regional migrations. Scientists have long been debating whether the origin

of what is considered "among the largest climate signal anywhere" [4] for the 20th century lied in the land-surface thermodynamics and land–atmosphere interactions [5,6], or rather in specific patterns of Sea Surface Temperatures (SST) in various oceanic basins [7–10]. Among the authors favouring the ocean as the main forcing factor, some have underlined the role of the Eastern Equatorial Atlantic on the interannual variability of the West African rainfall regime, with a warm (cold) summer SST anomaly favouring positive (negative) rainfall anomalies over the Guinea (and vice versa for the Sahel, e.g., [11,12]). More recently, a consensus seems to have emerged for primarily attributing this drought to a very specific pattern of SST in the various oceanic basins controlling the WAM dynamics (see e.g., [13]). It is, however, worth noting that the significant model biases in reproducing SST and rainfall modes of variability [14] leave space for other factors also playing a role, such as the regional scale land use changes and the modulation by the Saharan heat low [15–17] as emphasized following the AMMA campaigns [18].

Some of these factors would be reinforced in a warmer climate; any acceleration of this warming will affect the large-scale dynamics in the tropical belt as well as the monsoon system's dynamics at regional scale. Critically at stake are the competing roles of the direct green-house-gas-induced surface warming, particularly efficient over the Sahara Desert [19,20] and of the indirect global warming impact on SST [21]. In fact, while West Africa is often identified as a climate change "hot spot" due to the large projected changes in temperature and precipitation statistics [22], the expected regional climate evolution is far from clear and will affect various components of the rainfall regime such as the annual totals, the interannual to decadal variability, the intensities at small time scales, as well as both the spatial and temporal distributions of all these variables. Take rainfall intensification at daily to sub-daily time scales for instance as evidenced in the region by Panthou et al. [23,24] and Taylor et al. [20]: it is likely to be an early manifestation of the effect of global warming, as predicted by [25], with other regions in the world experiencing the same trend [26–28]. At small time scales, rainfall intensification may be closely related to the purely thermodynamical effect of the Clausius–Clapeyron law predicting an increase rate of $\sim$6.5%/$^\circ$K of the atmospheric moisture content. Hence, without change in the dynamics (from storm- to large-scale dynamics), and under constant relative humidity, extremes of precipitation at these temporal scales are expected to follow this rate, as suggested by e.g., Pall et al. [29]. On the other hand, Allen and Ingram [30] have shown, in a seminal study combining model simulations and observations, that a doubling in the atmospheric $CO_2$ concentration rather seems to yield a global mean annual precipitation increase of 3%/$^\circ$K, in relation to energy balance considerations at larger scales. On top of that, the regional system dynamics and its coupling with convection [31] will be affected by the differential heating along the Ocean-to-Sahara latitudinal transect. Thus, our capacity of predicting theoretically how the West African rainfall will be impacted by global warming over a range of space and time scales is inherently limited by our imperfect understanding of the interplay between various thermodynamical and dynamical factors (see e.g., [32] on this issue). Various modeling approaches have been used over the past 20 years in order to remedy, at least partly, this shortcoming.

*1.2. Large-Scale Dynamics Biases Versus Regional-Scale Physical Errors*

Gaining a finer understanding of how global warming will translate regionally is the new frontier of climate research. However, despite continuous improvements, General Circulation Models (GCMs) still fail at reproducing the regional monsoon systems, especially when it comes to the WAM. Apart from consistent results across the CMIP5 models regarding the reinforcement of the wet central Sahel-dry western Sahel dipole, GCMs do not position correctly the monsoon belt in latitude and they also disagree on the pattern of rainfall evolution in the context of global warming [33]. This caveat does not seem to be solved in the CMIP6 models; for instance, the Arpege Meteo-France model displays a deteriorated representation of the WAM as compared to the CMIP5 version [34]. Various factors are involved in the difficulty of GCMs to reproduce the monsoon systems: (i) the convective nature of precipitation which is not well taken into account, (ii) a mis-representation of ocean–atmosphere

interactions resulting in radiative biases and a wrong representation of low level oceanic cloud cover, especially near the coasts, (iii) a misrepresentation of surface–atmosphere interactions, leading to albedo and soil moisture biases, (iv) differing sensitivities to a given perturbation (e.g., an increase in GHGs concentration) inherited from each GCM particular design [1]. In this context, Regional Climate Models (RCMs) are commonly seen as valuable tools since their finer resolution allows for a better representation of regional to local small-scale processes that are not captured by GCMs [35]. The immediate advantage of the higher resolution is to allow for a better representation of surface conditions (topography, vegetation, oceanic eddies) that may play an important role in controlling the surface–atmosphere interactions and thus the regional climate dynamics. The CORDEX-Africa project was designed with the goal of improving climate simulations in Africa using multi-RCM ensemble runs. Rainfall regime climatology and interannual variability were indeed improved [36], but the ill positioning of the rainfall belt remains a major default.

There are two main factors explaining why RCMs are not solving all the problems encountered with GCMs. First, a finer resolution does not solve the caveats linked to the parameterization used in global climate models per se, most notably for convection, whose role is especially crucial in the formation of tropical precipitation. In this respect, the recent convection-permitting (CP) simulation over Africa presented in [37] is a real step forward. It improves the representation of the spatio-temporal distribution of rainfall and of the diurnal cycle; it also better captures the short-lived rainfall events. However, a bias remains in the latitudinal positioning of the rainfall belt due to the global 25-km resolution atmospheric model simulation used to force the CP model. This brings us to the second main limiting factor when using RCMs, linked to the propagation of errors persisting in the GCMs they are embedded in. Key biases in the representation of the tropical easterly jet and in the advection of humidity between the various tropical sub-regions (continents/oceans) thus remain, explaining why the rainfall belt remains positioned too far South. This problem of propagating errors inherited from the boundary forcing fields within the domain of RCM integration is a classical "garbage in, garbage out" issue [38]. Disentangling the role of the large-scale dynamics from that of the physical processes in an RCM misrepresentation of the monsoon dynamics is thus key for understanding and ranking the respective contributions of resolution, parametrisation and propagation of the boundary errors. Various approaches have been proposed to that end; one of them consists of using reanalyses as boundary forcing fields. Since reanalyses incorporate observations they are known for being less biased than unconstrained global climate models. Pan et al. [39], for instance, compare the results obtained in predicting rainfall patterns over the continental United States when using two different RCMs driven by either a reanalysis, a GCM under present climate or the same GCM run under a climate scenario at a $CO_2$ concentration of 480 ppm. They conclude that while boundary forcing related errors and inter-model errors are in the same order of magnitude—these errors depending on the season considered—both RCMs perform poorly in reproducing the observed rainfall pattern in present climate, whatever the type of forcing. They also show that the ratio of climate change to either boundary forcing biases or inter-model differences is substantially larger than 1, except in summer, thus associating regional climate projections with a degree of confidence. A different approach was recently taken by Diallo et al. [40] with the aim of better understanding the errors produced by the physical parameterization in a GCM by isolating them from dynamical biases. The horizontal winds of the GCM are nudged toward reanalysis, which allows to maintain the global coherency in temperature and humidity. Wind nudging is shown to greatly improve the location of the Inter-Tropical Convergence Zone (ITCZ) as well as the representation of the components of the surface energy budget directly impacted by the water budget and hence facilitates a more systematic analysis of remaining biases associated with the model physics. The surrogate approach presented below is another approach solely focusing on the effect of an atmospheric temperature and/or sea surface temperature increase within the simulation domain while using reanalyses as RCM driving data.

*1.3. A Surrogate Approach for Assessing a Regional Model Sensitivity to Boundary Forcing Fields Errors*

In order to reduce the uncertainty linked to GCM outputs while still allowing for assessing the effect of global warming over specific regions, the Surrogate Climate Change (SCC) methodology has been proposed by Schär et al. [41]. It consists of using reanalyses data as forcing fields of an RCM and performing air temperature and/or SST perturbations at the domain boundaries (the implicit associated assumptions are discussed in the next section). Van Lipzig et al. [42] use this methodology to study the sensitivity of the Antarctic ice sheet Surface Mass Balance (SMB) to changes in SST and sea ice conditions. Prescribing SST perturbations over a wide range of temperatures they conclude that the relationship linking the SMB to the low level moisture content might be more complex than usually thought. The SCC protocol is also used in the study of Im et al. [43], where a reduction in summer precipitation over the Alps is shown to be linked with a snow cover–soil moisture feedback mechanism. Using the Modèle Atmosphérique Régional (MAR) over Greenland, Delhasse et al. [44] evidence the strong impact of observed changes in the large-scale circulation not simulated by GCMs under future climate change scenario (namely blocking situations) on the ice sheet surface mass balance in a warmer world. Hence, this experimental protocol has proven instructive to studies dedicated to the global warming influence on the climate of mid- and high-latitude regions.

The objective of this study (to our knowledge the first of this kind for West Africa) is two-fold: understanding changes in the West African rainfall regime owing to thermodynamical perturbations (large-scale circulation fields being prescribed at the lateral boundaries) and assessing the model sensitivity to errors in the boundary forcing fields. Ultimately, this should help to better understand the model functioning and allow for a more informed use of its outputs under future climate change scenarios. The model and experimental protocol are described in Section 2. Results of the model performances evaluation with respect to three independent datasets are given in Section 3, together with the results of the sensitivity experiments. The main results of this study and the associated limitations are given in Section 4.

## 2. Materials and Methods

*2.1. Model Description*

The RCM used in this study is the Modèle Atmosphérique Régional (MAR), a Limited Area atmospheric model resolving the hydrostatic primitive equations on a three-dimensional grid using the full continuity equation. The dry dynamics of the model are thoroughly described in Gallée and Schayes [45] and the cloud micro-physical processes, based on the parameterization by Kessler [46] and Lin et al. [47], are described in Gallée [48]. The model was initially developed for polar regions and then adapted to the tropics by coupling it to a mass-flux adjustment convection scheme based on the work of Bechtold et al. [49]. The model has 40 vertical levels with a terrain-following normalized ($\sigma$) pressure coordinate offering a proper representation of topography. Vertical resolution decreases with height for a better representation of low level processes. The MAR is coupled to a one dimensional soil–vegetation model (Surface Vegetation Atmosphere Transfer, SVAT, [50]) containing one vegetation layer and 7 soil layers whose thickness increases with depth, allowing for a more realistic representation of surface and soil properties and processes. Each grid cell can have three different vegetation types in this study, from which energy and turbulent fluxes are computed separately and averaged using weighting coefficients corresponding to the fraction of each vegetation type. Interactive land surface schemes, ensuring a more accurate balance between precipitation, evapotranspiration, runoff and soil moisture are recognized to improve model performances [51]. The calculation of turbulent fluxes are based on the Monin-Obukhov similarity theory in the surface boundary layer and on the k-$\epsilon$ model of Duynkerke [52] above. The radiative scheme is that of Morcrette [53]. A 3-minutes time step is used for the computation of dynamics.

The MAR was first used over West Africa to evaluate its ability in properly simulating the WAM rainfall regime [54]. It was shown to reproduce the intraseasonal variability of rainfall together with the abrupt northward shift of the rain-band, i.e., the transition between the oceanic and continental regimes [55]. This feature, referred to as the "monsoon jump", was the subject of a study by Ramel et al. [56], where it was shown that the northward migration of the surface heating maximum from the sudano-guinean region to the Sahara in mid-June is at least partly responsible for the northward migration of the rain band over the Sahel. The MAR was also used to study the influence of the Equatorial Atlantic Ocean surface temperature anomalies on the West Africa rainfall regime [12]. This study shows the dominant role of the SST on the 1983–1984 interannual variability of the WAM.

### 2.2. Experimental Protocol

### 2.2.1. Domain and Input Data

The integration domain (Figure 1) is chosen large enough (25° W–21° E, 10° S–40° N) for the atmospheric processes to adjust to the topography and the water surfaces of the Atlantic Ocean, the Mediterranean sea and the Chad Lake. The region over which it can be assumed that the flow has adjusted to the MAR grid, referred to in the following as the "West Africa" (WA) study domain, extends from 0° to 30° N and 18° W to 18° E. Our study will also focus on two inner regions, namely the Guinea (18° W–8° E, 4° N-11° N) and the Sahel (18° W–8° E, 11° N–18° N).

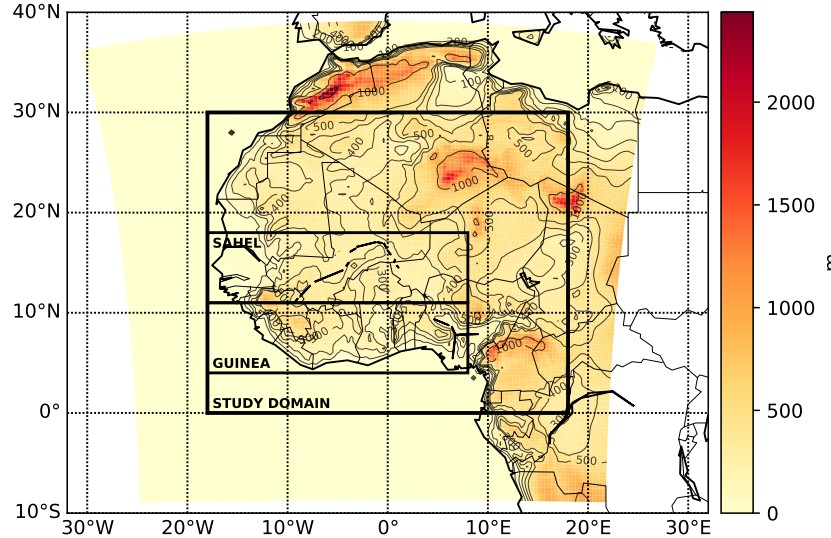

**Figure 1.** MAR integration domain and model topography (shading), the West Afica study domain, Guinea and Sahel boxes.

A 40 km horizontal grid spacing is used; a trade-off between a good enough representation of small scale atmospheric processes and computational costs. It is assumed that a higher resolution would not lead to significant improvement of the model performances with respect to the objectives of this study, as the hydrostatic approximation does not allow for the large vertical accelerations that would occur at a higher spatial resolution due to convection. The lateral (atmospheric) and lower (oceanic) boundary conditions are provided by the ERA-interim reanalyses [57] with a 6 hour time step. The vegetation cover is prescribed by the Leaf Area Index (LAI) from the MERRA-2 reanalyses [58] on a monthly basis to ensure a proper representation of its seasonal cycle. The forcing fields have a lower spatial resolution and are then interpolated on the model grid. The gain from dynamically downscaling reanalyses with the MAR along with its sensitivity to different boundary forcing origins and resolutions is not the purpose of our study.

2.2.2. Study Period

The simulations are performed on the following years: 1983, 1984, 1993, 1994, 2010 and 2011, chosen for their contrasted rainfall patterns; 1983–1984 are in the core of the drought period, with a larger dry anomaly in 1983 over the Guinea and a rainfall dipole (wet Guinea/dry Sahel) in 1984. The years 1993 and 1994 are closer to the study period average with 1993 a slightly dry and 1994 a wet year (both anomalies are particularly prominent over the Sahel). Finally, 2010 is a very wet year whereas 2011 is close to the study period average with a reversed rainfall dipole (dry Guinea/wet Sahel). The extent to which the model is able to capture this interannual variability is assessed in the next section. The sampled years then cover different modes of variability: decadal when couples of years are compared with each other and interannual within each couple of years. This provides robustness to our results, as it allows assessing the importance of the thermodynamical perturbations relative to the internal variability of the climate system. Finally, a one-year spin-up is performed prior to each 2-years period for the soil conditions to adjust [59].

2.2.3. Protocol

Our experiments follow the Surrogate Climate Change (SCC) framework as introduced by Schär et al. [41]: boundary forcing fields are prescribed from reanalyses and homogeneous air temperature and/or SST perturbations are added at the integration domain boundaries, leaving the RCM free to adjust its internal thermodynamics and dynamics to these changed forcing conditions. Hence, the boundary forcing perturbations can be isolated as responsible for the changes in the model behaviour. The experimental design is as follows:

- T00: control simulation
- T10: air temperature increase of 1 °C over the whole atmospheric column,
- T01: horizontally homogeneous SST increase of 1 °C,
- T11: a combination of the two previous perturbations.

In addition, the relative humidity is kept constant for all experiments, following the expectation of a constant relative humidity with global warming [60]. The result is an increase in atmospheric moisture content of $\sim$6.5%/°K in the range of present air temperatures according to the Clausius–Clapeyron relationship. The first experiment (T10) is assumed consistent with a widespread tropospheric warming while the second one (T01) corresponds to a warmer regional SST. The third experiment (T11) aims at comparing the relative impact of each perturbation and to test for a possible synergetic effect. This experimental protocol has several limitations. First, the prescribed boundary forcing perturbations would not be realistic on the long-term range, since an atmosphere–ocean equilibrium might be reached at some point. Second, several aspects of the climate system are not taken into account: changes in vegetation, ocean dynamics and green-house gases concentrations. However, as is the case for the atmosphere–ocean coupling, interactions involving these components occur on longer time scales compared with the length of our simulations. Thus, these are not limiting points for the sensitivity analysis performed in this study. Note that the prescribed large-scale atmospheric circulation at the domain boundaries ensures a consistent circulation, i.e., not relying on GCM outputs and associated uncertainties. Finally, a critical point when applying such perturbations is to create a clear response while ensuring the physical consistency of the WAM, which turns out to be the case.

## 3. Results

*3.1. Model Evaluation*

Before using the MAR for SCC experiments its ability to reproduce the main features of the WAM is assessed using three independent datasets:

- CHIRPS [61], an observationally constrained satellite product available since 1982 covering the study period,

- BADOPLU (BAse de DOnnées PLUviomètres), a rain-gauge product gathering since 1950 in-situ observations from various national meteorological agencies in a fully quality-controlled dataset (see the supplementary materials of Panthou et al. [24] for a detailed description of the data processing). The point rainfall data from BADOPLU are spatially interpolated on a $1 \times 1°$ regular grid by a block-kriging technique using a double exponential structure variogram (see [62] for details of the interpolation),

- ERA5 [63], the new global atmospheric reanalysis produced by the ECMWF, spanning the period from 1979 to present with a $0.25 \times 0.25°$ grid spacing. Note that since ERA5 data were collected for a domain extending from 20° W–20° E and 0°–20° N (from the Copernicus Climate Change Service portal: https://cds.climate.copernicus.eu/cdsapp#!/home), this reduced window is considered for the model evaluation.

### 3.1.1. Spatial Pattern

The spatial distribution of the simulated seasonal cumulative precipitation is first evaluated. The emphasis is on the July -August–September (JAS) period, the core of the rainy season in the Sahel. Figure 2 shows the JAS total rainfall amounts averaged over the six years of the study period for the MAR and the three comparative datasets.

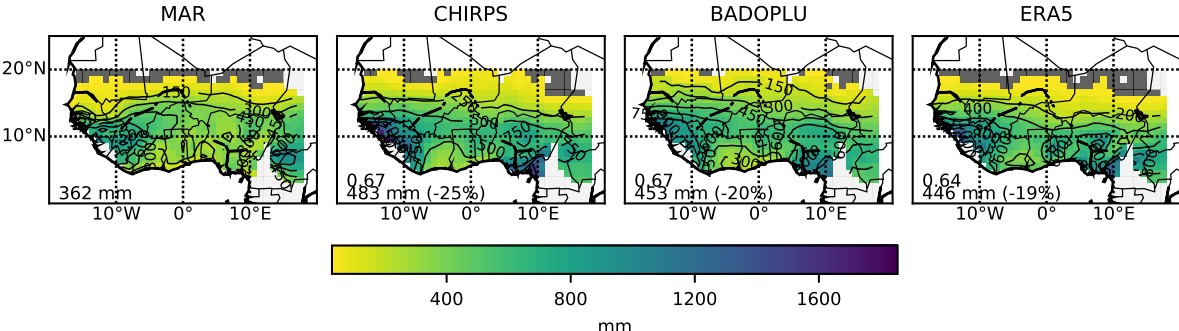

**Figure 2.** Study period averages of JAS total rainfall for MAR, CHIRPS, BADOPLU and ERA5 datasets. Grid cells with JAS rainfall amounts < 30 mm are ignored (dark grey shading), removing 11.0%, 7.3%, 3.2% and 15.5% of the MAR, CHIRPS, BADOPLU and ERA5 JAS total rainfall, respectively. Grid cells with missing value for at least one dataset are also ignored (light grey shading). Values in the bottom left corner are study domain mean values (mm), model anomalies relative to the comparative dataset (%) and spatial coefficient of correlation between the model outputs and each comparative dataset.

The model reproduces the latitudinal rainfall gradient and the two peak rainfall areas associated with mountain ranges, namely the Fouta Djalon and Mount Cameroun. However, the latter is too far inland in the model, likely the result of a reduced model topography (Mount Cameroun is only ~1500 m high in the model). Over the WA study domain the MAR is 25%, 20% and 19% drier compared with the satellite, the rain-gauge and the reanalysis products, respectively. This dry bias might in part be related to the generation of too many small rainfall events (not shown), a common feature of climate models [25]. The spatial agreement between the model and the comparative datasets, evaluated with a Spearman's rank coefficient of correlation (i.e., assessing a monotonic relationship between two variables), is satisfying, with values of 0.67 with respect to CHIRPS and BADOPU and 0.64 for ERA5. The absolute and relative biases of the model outputs with respect to each comparative dataset for WA, Guinea and Sahel regions, together with the spatial coefficients of correlation are summarized in Table 1. Note that only grid cell with values for the four datasets are considered for this evaluation. Considering separately Guinea and Sahel, the model performances display larger dry biases, showing the difficulty to properly capture regional scale features of the WAM. Worth noticing is the spatial agreement that decreases over Guinea (due to the two under-estimated peak rainfall

areas) but increases over the Sahel, evidencing the model ability to faithfully simulate the main rainfall spatial pattern over this region (i.e., the meridional gradient).

**Table 1.** Study period averages of JAS cumulative rainfall (mm) for MAR, CHIRPS, BADOPLU and ERA5. The model anomalies with respect to each comparative dataset are expressed in absolute and relative terms. The spatial correlation is calculated with a Spearman coefficient of correlation.

| Datasets | | WA | Guinea | Sahel |
|---|---|---|---|---|
| MAR (T00) | mean | 362 | 428 | 299 |
| CHIRPS | mean | 483 | 672 | 415 |
| | difference | −113 | −244 | −116 |
| | % change | −25 | −36 | −28 |
| | spatial c | 0.67 | 0.63 | 0.75 |
| BADOPLU | mean | 453 | 575 | 438 |
| | difference | −91 | −147 | −139 |
| | % change | −20 | −26 | −32 |
| | spatial c | 0.67 | 0.59 | 0.71 |
| ERA5 | mean | 446 | 650 | 315 |
| | difference | −84 | −222 | −16 |
| | % change | −19 | −34 | −5 |
| | spatial c | 0.64 | 0.52 | 0.72 |

### 3.1.2. Seasonal Cycle

The temporal characteristics of the simulated rainfall regime are next evaluated by comparing the simulated seasonal cycle, averaged over the study period, with those of the comparative datasets (Figure 3). Note that the daily rainfall signal is smoothed out with a 10-days running mean to filter out the high-frequency variability. The JAS dry bias is present over the two regions with the MAR displaying lower daily rainfall amounts over the whole rainy season, with the exception of a wetter early summer (June–July) in the MAR compared to ERA5. The main modes of variability are partly captured by the model with a faithful mono-modal shape over Sahel and a poorer bi-modal shape over Guinea. Over this region, the dry season (November, December, January, February, March) is wetter in the model. Also of importance here is the synoptic-range variability (∼10 days), in good agreement over the two regions of interest, showing the model ability to simulate the monsoon dynamics at this time scale.

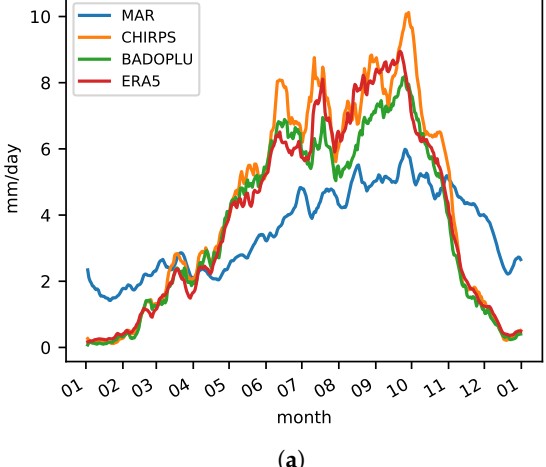

(**a**)

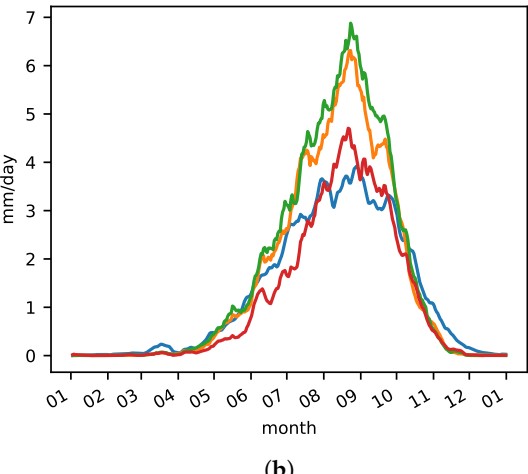

**(b)**

**Figure 3.** (**a**) Guinea and (**b**) Sahel daily rainfall time series with a 10-days running mean for the model (blue), satellite (orange) and rain-gauge (green) datasets.

### 3.1.3. Interannual Variability

The model ability to capture the interannual variability over the study period is finally assessed. The JAS cumulative rainfall departures (i.e., relative differences) from the study period average are computed for each dataset (Figure 4). The interannual variability is better captured over WA than over the Guinea and Sahel regions, consistent with the previously stated difficulty in capturing specific regional features of the WAM. Of interest is the contrasted performances for 1984 and 1994. These years display a dipole-like rainfall pattern of opposite sign, i.e., a wet Guinea/dry Sahel (1984) and a dry Guinea/wet Sahel (1994), linked with warm and cold SST anomalies in the Gulf of Guinea, respectively [14]. The former is well simulated, with both rainfall anomalies captured, while the latter is not, with none of the anomalies captured. Hence, all other things equal, the model looks more sensitive to warm SST anomalies, consistent with the work of Messager et al. [12]. More specifically, the influence of a warm SST anomaly on the regional patterns of rainfall (wet Guinea/dry Sahel) is well captured while the cold SST influence (dry Guinea/wet Sahel) is not. This is likely due to the more complex processes involved in the cold-SST rainfall dipole, as rainfall must be sufficiently limited on Guinea for enough moisture to be supplied and eventually precipitated out over the Sahel. To finish, the recent period (2010, 2011) interannual variability is well captured, in both sign and magnitude, over the three regions considered.

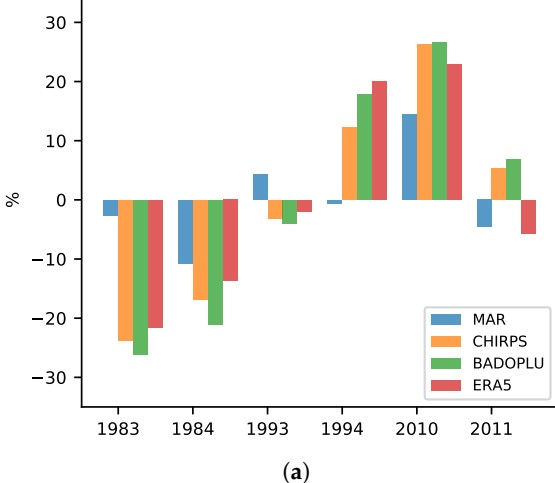

**(a)**

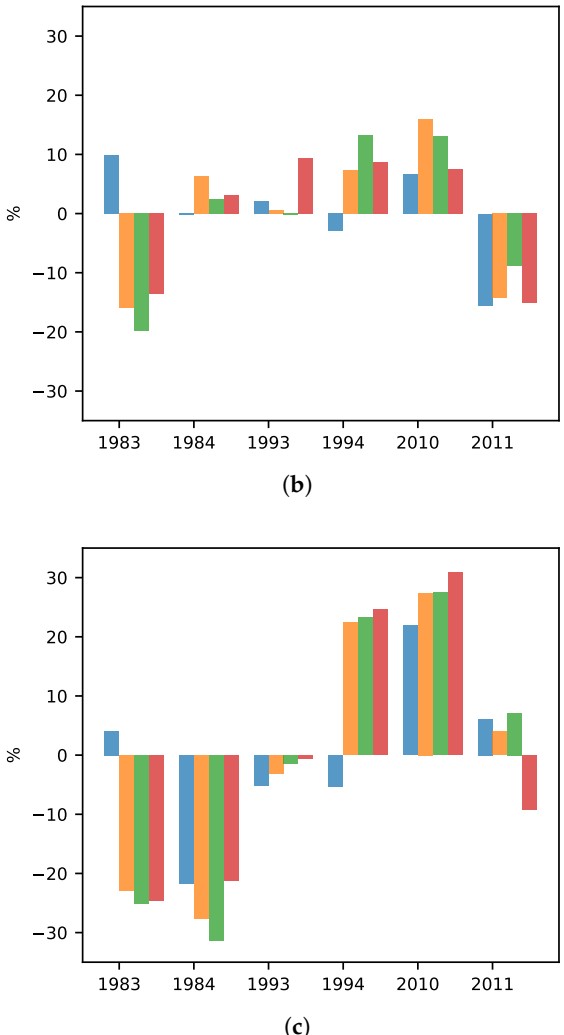

**Figure 4.** (**a**) WA, (**b**) Guinea and (**c**) Sahel JAS cumulative rainfall departures from each dataset study period average (%).

### 3.2. Sensitivity Experiment Results

#### 3.2.1. Rainfall Response

The objective of this section is to analyze the main features of the model response to the boundary forcing perturbations, with a special focus on the JAS cumulative rainfall anomalies. Seasonal- and regional-scale changes in the simulated WAM thermodynamics and dynamics are then considered.

The Figure 5 shows the difference in the spatial mean of JAS cumulative rainfall between each sensitivity experiment and the control simulation, averaged over the study period. First, T10 and T01 display contrasted patterns of JAS rainfall anomalies: the air temperature warming (T10) results in a widespread drying tendency of −22.3 mm (−14.3%) while the warm SST perturbation (T01) yields a wetting tendency of +27.1 mm (+17.4%). The combination of the two perturbations (T11) results in an overall wetting tendency of 7.95 mm (+5.1%). These study domain mean rainfall anomalies hide regional discrepancies, as shown in Table 2. First, the rainfall anomalies are of larger magnitude over the Guinea than the Sahel, consistent with the statement of Cook and Vizy [14] that the drying tendency over the Sahel during wet Guinea/dry Sahel events in response to warm SST anomalies is harder to capture. Similarly, the western Sahel exhibits larger rainfall anomalies than the eastern Sahel. These regional discrepancies are indicative of the more complex dynamics involved in the rainfall response further inland. Also worth noting is the differing behaviours of the T11 experiment response:

over Guinea the response is quasi additive with the larger T01 wetting tendency (+30.9%) dominating over the T10 drying tendency (−22.7%). This is not the case over the Sahel, where a stronger drying in T10 (−7.3%) combined with a smaller wetting in T01 (+3.7%) yields a wet anomaly in T11 (+5.3%). This non-additive response also holds for the western and eastern Sahel areas. As already mentioned, this sahelian dipole pattern of rainfall anomalies is an expected feature of the future rainfall change in West Africa. However, for succintness, only the main features of the rainfall response are analyzed.

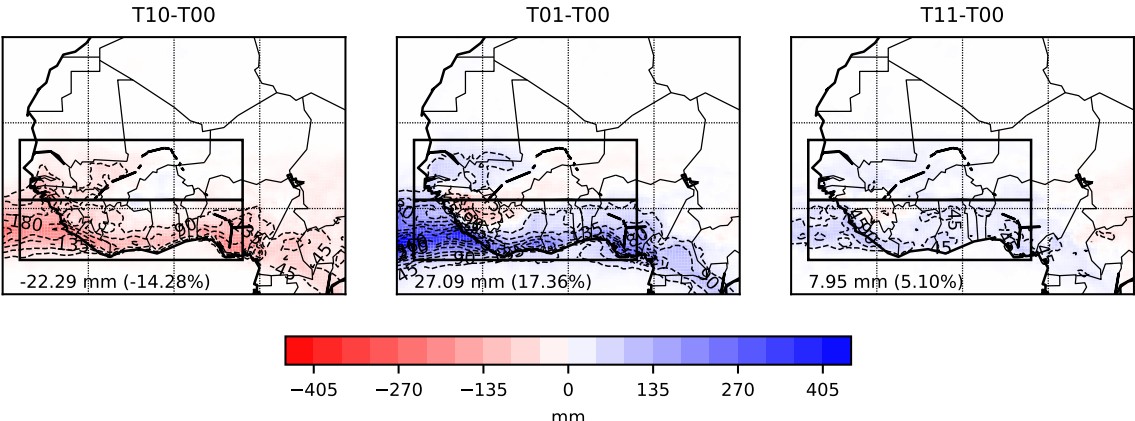

**Figure 5.** JAS cumulative rainfall anomalies for the three sensitivity experiments with respect to the control simulation averaged over the study period. Numbers in the bottom left corner are study domain (WA) mean absolute and relative (in parenthesis) anomalies.

**Table 2.** JAS cumulative rainfall (mm) for the control simulation (T00) and sensitivity experiment anomalies with respect to the control simulation. Relative anomalies (%) are displayed in parenthesis.

| Anomalies | Guinea | Sahel | W Sahel | E Sahel |
|---|---|---|---|---|
| T00 | 428 | 299 | 282 | 314 |
| T10-T00 | −75.2 (−22.7) | −20.6 (−7.3) | −37.9 (−15.3) | −2.2 (−0.7) |
| T01-T00 | 102.4 (30.9) | 10.3 (3.7) | 26.0 (10.5) | −6.3 (−2.0) |
| T11-T00 | 33.1 (10.0) | 14.8 (5.3) | 17.2 (6.9) | 12.4 (3.9) |

In order to assess how robust this signal is, the interannual variability of the JAS rainfall anomalies with respect to the control simulation study period average is investigated (Figure 6). Yearly values of JAS rainfall anomalies for each sensitivity experiment with respect to the control simulation values are summarized in Table A1 (Appendix A).

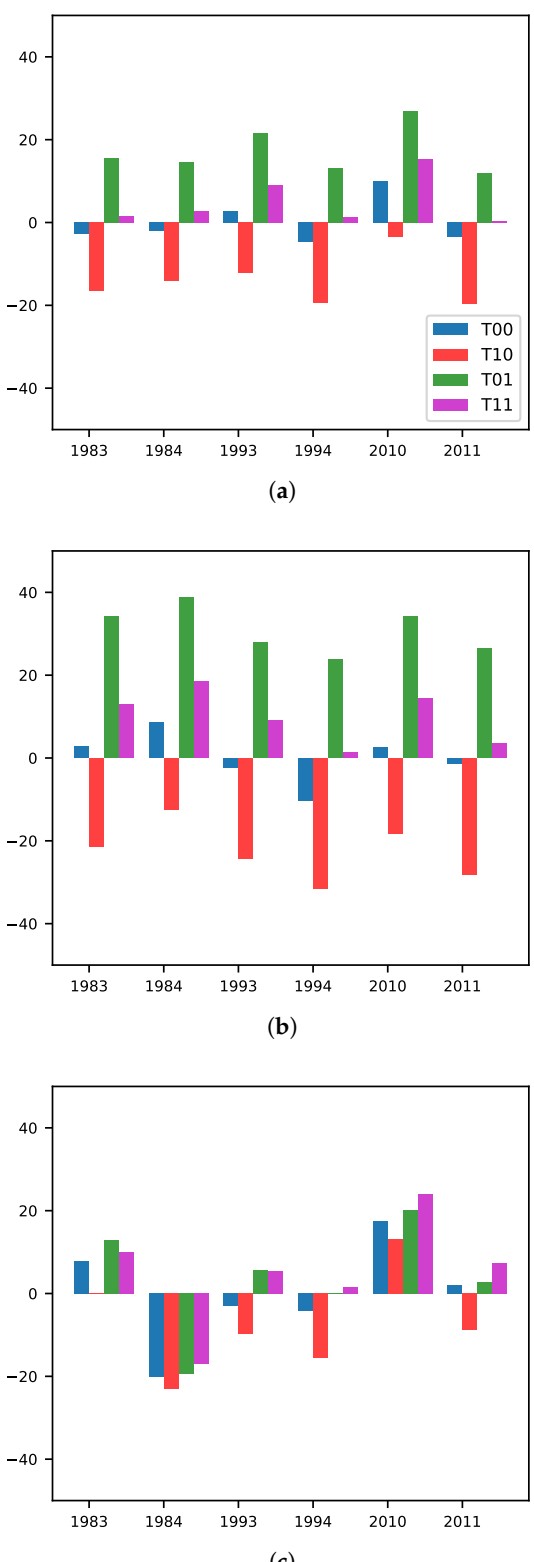

**Figure 6.** (**a**) WA, (**b**) Guinea and (**c**) Sahel interannual variability of the JAS rainfall anomalies relative to the control simulation study period average.

Two main conclusions can be drawn:

- the rainfall response to the perturbed boundary forcing over the regions of interest is unequivocal, with T10 on the one side and T01 and T11 on the other side *always* displaying dry and wet anomalies, respectively,
- over the WA domain and the Guinea region, the JAS rainfall changes are beyond the range of natural variability, defined by the interannual variability in the control simulation (blue bars in Figure 6). This feature indicates the strong model sensitivity to thermodynamical perturbations over this region and is suggestive of a dominant mechanism shaping the rainfall response. Over the Sahel, the rainfall anomalies relative to the study period average have the same amplitude as the control simulation interannual variability. Therefore, the dynamical influence (i.e., the year-to-year internal variability) on the boundary perturbation sensitivity may be larger in this region than over Guinea. Note here the added value of the sampled years, representative of distinct climatic conditions in West Africa, adding robustness to these conclusions.

In Section 3.1, the MAR was shown to display a dry bias over the Guinea of 36%, 26% and 34% with respect to the satellite, the rain-gauge and the reanalysis datasets, respectively (Table 1). When considering rainfall anomalies over land only (as in the model evaluation) the T10, T01 and T11 experiments display rainfall anomalies of −16%, +14% and +7%, respectively, over the Guinea region. Hence, these anomalies compare relatively well with the model intrinsic bias. This comparison is less prominent over the Sahel. However, the model intrinsic bias has a dipole structure over this region, with a drier western/wetter eastern Sahel pattern (not shown). The T01 experiment displays a clear dipole structure of its rainfall response over this region, with a wetting (drying) tendency over the western (eastern) Sahel (see Table 2). As a result, both the overall dry bias and the spatial agreement are improved in the T01 experiment, when compared to the comparative datasets: the former changes from −25%, −20% and −19% to −19%, −14% and −13% with respect to CHIRPS, BADOPLU and ERA5 respectively, while the latter reaches ∼0.75 instead of ∼0.65 in the control simulation (see Figure A1 in Appendix B). Moreover, the experimental perturbations prescribed to the boundary forcing fields, particularly the SST one, are within the range of current GCM biases [64], giving relevance for such a sensitivity analysis. It is also worth mentioning is the fact that both the warmer SST alone (T01) and combined with a 1K-warmer atmosphere (T11) have a sizeable impact on the simulated West African rainfall regime, highlighting the crucial need for the atmosphere–ocean coupling to be correctly captured by AOGCM driving higher-resolution models.

### 3.2.2. Physical Interpretation

In this section, the model responses to the boundary forcing perturbations are analyzed, first through several atmospheric variables and then using the Moist Static Energy (MSE) framework. Changes in dynamical aspects are finally reviewed. Note that if the limited length of the simulations does not allow for a statistical inference of cause–effect relationships, such an analysis might provide clues on the mechanisms involved in the rainfall regime sensitivity, together with some confidence on the physical consistency of the model response. Note that only regional mean values are dealt with, although it may not be fully representative of the spatial variability, particularly over the Sahel due to the dipole structure of the rainfall response (and possibly of the associated mechanisms). However, a thorough disentangling of the mechanisms involved in the rainfall response is not within the scope of this study.

(i)　Thermodynamics

First, changes in the seasonal- and regional-mean vertical profiles of temperature, specific humidity and relative humidity are analyzed (Figure 7). The vertically homogeneous air temperature warming at the lateral boundaries (T10) stabilizes the vertical temperature profile, with a larger warming in the upper troposphere. This is likely the consequence of the prescribed unchanged

SST that limits the temperature increase at low levels. As a result, the temperature stabilization is larger over the Guinea region (Figure 7a). Conversely, the warmer SST (T01) destabilizes the vertical temperature profile, with a larger temperature increase in the low troposphere, more prominent over Guinea than the Sahel. In T11, both effects combine: the air temperature increase is larger at 925 hPa and in the upper troposphere than in the mid-troposphere, resulting in a smaller destabilization.

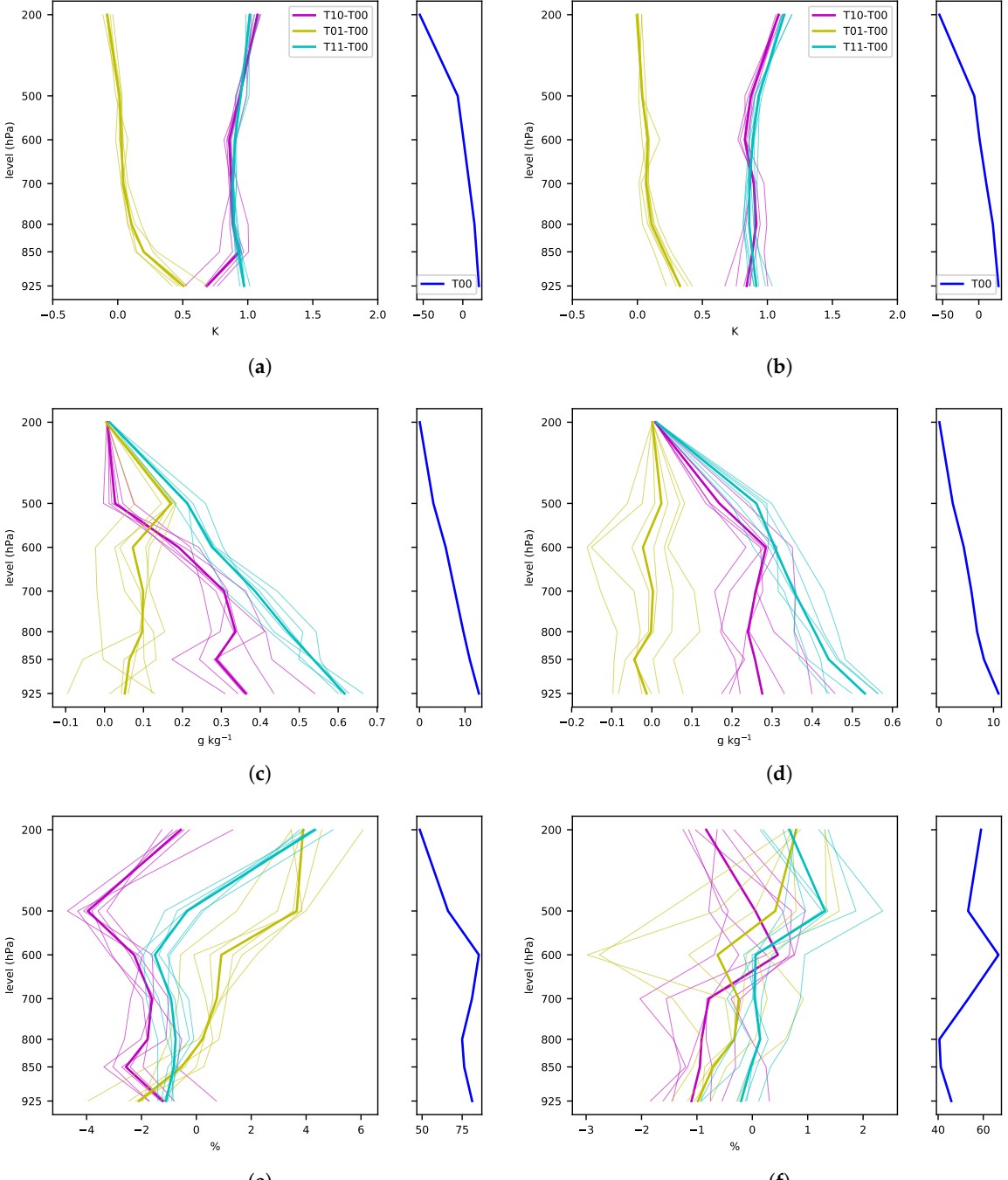

**Figure 7.** Mean JAS vertical profiles of air temperature (**a**, **b**), specific (**c**, **d**) and relative (**e**, **f**) humidity anomalies for Guinea (**a**, **c**, **e**) and Sahel (**b**, **d**, **e**). The reference vertical profile of each variable is shown on the right of each plot. Thick lines are study period medians and thin lines are individual years.

The main difference between T01 and T11 relates to the changes in the atmospheric moisture content. In T01, it only slightly increases over the Guinea (Figure 7c) while it is reduced ove the Sahel (Figure 7d). This limited increase in the lower troposphere moisture content in T01 is somehow

counter-intuitive, as one would expect a warmer SST to result in an enhanced evaporation and, subsequently, a moister lower troposphere. It is shown in the next part that this feature might be linked to a dynamical mechanism. In T11 both regions undergo large increases in moisture content, as is the case in T10 too, yet to a smaller extent. However, the vertical profiles of relative humidity (Figure 7e,f) show that in this latter experiment, the troposphere is indeed drier over the two regions. Over the Guinea, the warm SST perturbation results in a drying at 925 hPa and a substantial moistening in the mid- and upper-troposphere. Over the Sahel, the T01 and T11 experiments share a similar pattern: the relative humidity is reduced in the lower troposphere and increased above ∼600 hPa. This analysis shows that on a regional scale, the increased moisture holding capacity of the atmosphere only results in a relatively more moist atmosphere when sufficient additional moisture is supplied through evaporation over the ocean, that is, when both perturbations are combined. Note the much larger spread among individual years (thin lines in Figure 7) of these vertical profiles over the Sahel, indicating a larger contribution of the internal variability to the anomaly patterns.

The temperature and moisture anomalies also display strong spatial variability in their horizontal distributions. An illustration is given by the change in specific humidity at 925 hPa, i.e., where the largest climatological values are found (Figure 8). Specific humidity anomalies in the sensitivity experiments are also the largest (in absolute terms) at this level (exept in T01, Figure 7c,d). Despite a smaller regional mean increase, T01 displays more spatial variability. The largest increase is located near the Equator and decreases as latitude increases. The specific humidity decrease off the Guinea coast around 10° N is associated with a strong warming (Figure A2 in Appendix C), resulting in a drop in relative humidity (Figure A3 in Appendix D), and is compensated for at upper levels, likely the effect on an increased convective activity in this region.

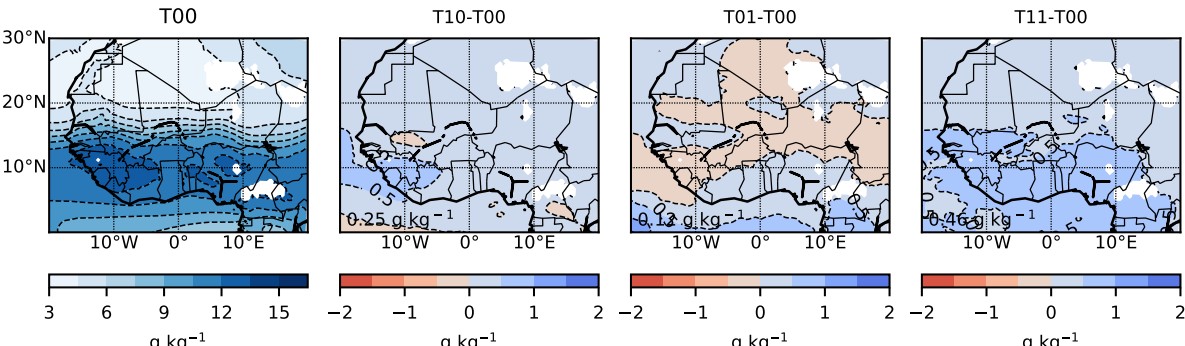

**Figure 8.** Mean JAS 925 hPa specific humidity for the control simulation and sensitivity experiment anomalies. Values in the bottom left corner are study domain averages.

These aforementioned thermodynamical changes bring us to analyze the changes in vertical stability in order to shed more additional light on the changes in rainfall. The MSE framework, first introduced by Neelin and Held [65], relates key thermodynamical and dynamical components to the atmospheric energy budget and the water cycle. Here, the changes in MSE and its vertical gradient, namely, the Moist Static Stability (MSS), are analyzed with a view to understand the precipitation changes shown in Figure 5. The MSE measures the total energy contained in an air parcel, exept kinetic energy. It sums up dry air enthalpy ($c_p T$), energy associated with the vaporization/condensation of water ($Lq$) and potential energy ($gz$):

$$MSE = c_p T + Lq + gz \qquad (1)$$

where $c_p$ is the heat capacity per unit mass of dry air (1004 J K$^{-1}$ kg$^{-1}$) at constant pressure, $T$ the temperature (K), $L$ the latent heat of vaporisation (2500 kJ kg$^{-1}$, irrespective of its negligible temperature dependence), $q$ the specific humidity (g kg$^{-1}$), $g$ the acceleration of gravity (9.81 m s$^{-2}$) and $z$ the altitude (m). The MSS corresponds to the vertical gradient of MSE, with positive MSS

(in pressure units) indicating instability. Vertical MSE and MSS profiles for each experiment as well as sensitivity experiment anomalies for these two quantities are displayed in Figure 9.

As mentioned in the previous part, changes are of larger magnitude over Guinea than Sahel, where the spread of vertical MSE and MSS anomaly profiles is also much larger. Therefore, the following analysis focuses on the Guinea region, although conclusions might also hold for the Sahel. In T10 the instability increases between 800 and 500 hPa (positive MSS anomaly) whereas stability increases below 850 hPa and with a larger magnitude above 500 hPa (negative MSS anomalies). T01 displays an opposite pattern of change, with positive MSS anomaly between 925 and 800 hPa and above 600 hPa. Finally, T11 displays a positive MSS anomaly over the whole troposphere, yet with a smaller magnitude than T01 below 850 hPa and above 500 hPa. Such changes in vertical stability are consistent with the rainfall anomaly patterns evidenced in the previous section: the overall stability increase in T10 is linked with negative rainfall anomaly whereas the increased instability in T01, and to a lesser extent in T11, are linked with positive rainfall anomalies over the Guinea region. These results are also consistent with that of Giannini [66], where the remote, global ocean warming-driven stabilization results in convective inhibition over tropical regions, while a local destabilization occurs through increased evaporation (driven by radiative forcing and enhanced by water-vapour feedback) and the subsequent increase in MSE at low levels. Analyzing in more details the contributions of each term, temperature changes at low levels drive the stabilization and destabilization in T10 and T01, respectively, while the increase in moisture has a larger contribution to the overall MSS decrease in T11 (see Figures A4 and A5).

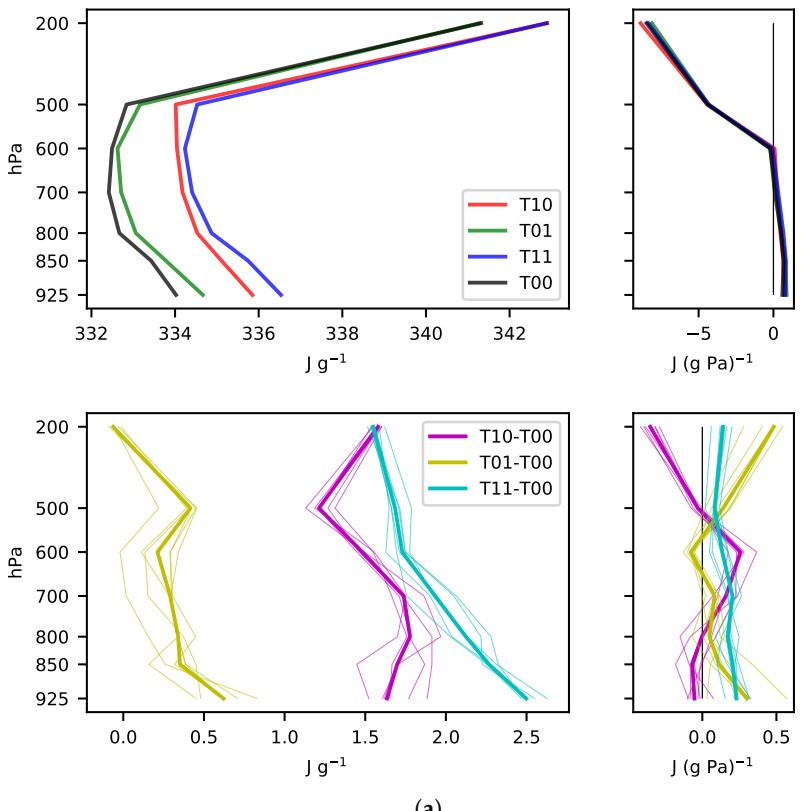

(**a**)

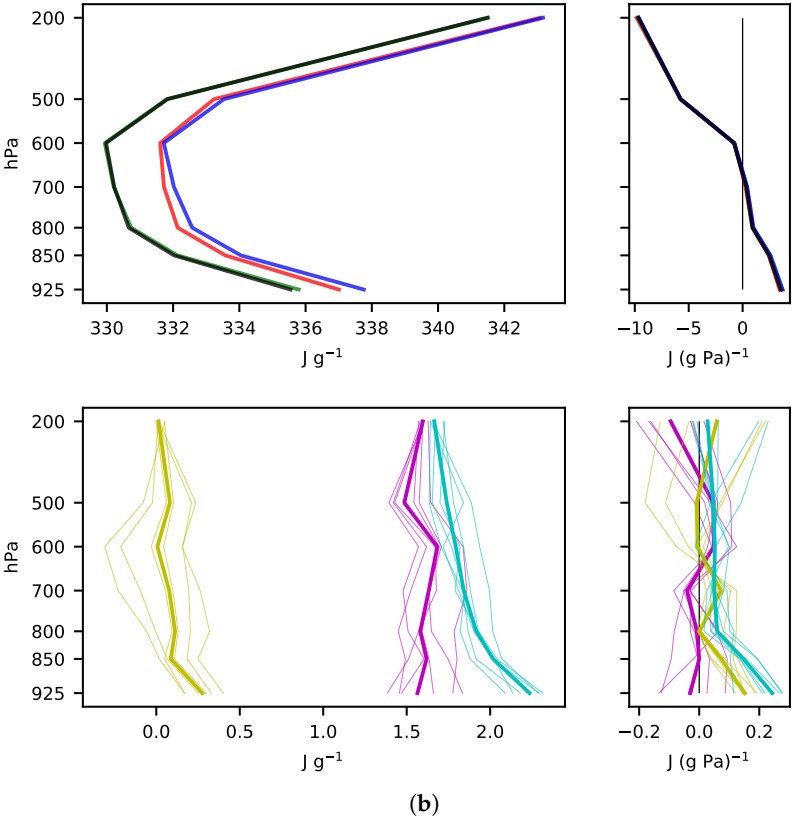

**Figure 9.** Guinea (**a**) and Sahel (**b**) regional mean JAS vertical profile of MSE (**left**) and MSS (**right**) for the control and sensitivity experiments (**top**) and anomalies (**bottom**). Thick lines are study period median values and thin lines are individual years.

(ii)   Dynamics

Changes in the MSE content and more specifically of its vertical and meridional gradients might affect vertical as well as horizontal motions, respectively. The Figure 10 shows the regional mean changes in vertical wind velocity between the control simulation and the sensitivity experiments over the Guinea and the Sahel. Over both regions T10 and T01 are associated with negative and positive anomalies, respectively, over the whole tropospheric column. Increased ascent is the telltale of an increased convective activity, linked with positive rainfall anomaly. Conversely, negative vertical motion anomalies in a region of ascent indicates a reduced convective activity. Hence, vertical motion changes are consistent with the rainfall anomalies over both regions and in both experiments. Note that not only the sign but also the magnitude of these anomalies are consistent with the rainfall changes, with a larger (smaller) magnitude of increase (decrease) over Guinea (Sahel). Interestingly, in T01, the increased ascent is largest in the mid-troposphere, where the largest specific humidity increase also stands, especially over Guinea (Figure 7c).

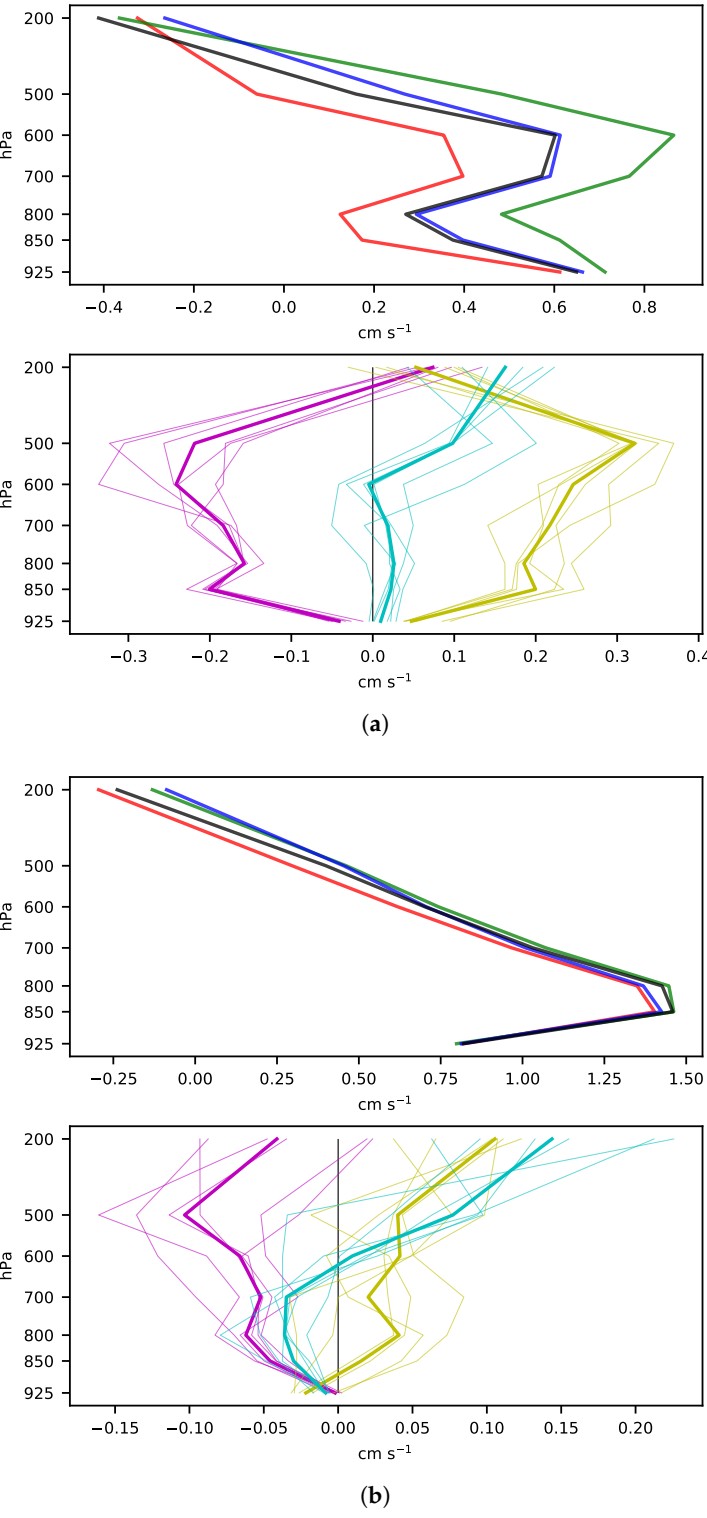

**Figure 10.** Guinea (**a**) and Sahel (**b**) regional mean JAS vertical wind speed for the control and sensitivity experiments (**top**) and anomalies (**bottom**). The velocity units is in cm s$^{-1}$ meaning that positive (negative) values are oriented upward (downward). Thick lines are study period median values and thin lines are individual years.

To finish, Figure 11 shows the 925 hPa horizontal wind speed and associated moisture transport in the control simulation together with the sensitivity experiment anomalies. In T10, the southerly monsoon flow is reduced over the Gulf of Guinea. This slowing down is only partly compensated for by the specific humidity increase (Figure 8) on the Guinea Coast, resulting in a slight increase in moisture transport. However, the larger horizontal wind speed decrease in the western part of the Gulf of Guinea results in a strong moisture transport reduction in this area, in spite of an increase in specific humidity. This feature is coincident with the location of strong rainfall reduction (Figure 5). Changes in moisture transport are larger in T01, with a strong increase near the Equator as a result of both a stronger and more northerly horizontal wind and a moister boundary layer. Over the Sahel, the moisture transport is reduced, mainly as a consequence of the drying of the lower tropospheric levels (Figures 7d and 8). In T11, changes in the low level horizontal wind speed are small. Therefore, the slight moisture transport increase is mostly the result of the widespread specific moisture increase at this level (Figure 8).

A feature worth noticing is the circulation anomalies that develop over the western Sahel: anticyclonic in T10 and cyclonic in T01, resulting in northerly (southerly) advection of dry (wet) air over this region. Wether this is a response to, or the cause of the reduced (increased) convective activity in T10 (T01) over this region would require more investigations, as would a more thorough analysis of the complex thermodynamical-dynamical interactions in the region.

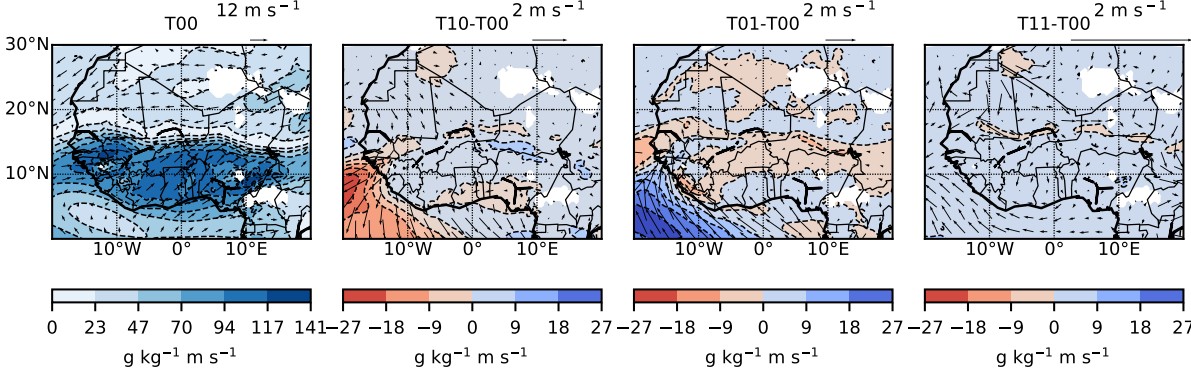

**Figure 11.** Mean JAS 925 hPa wind speed (arrows) and moisture transport (shading) for the control simulation and sensitivity experiment anomalies. Note the different wind speed scales.

In this part, changes in the WAM dynamics have been explored. These changes appear consistent with the rainfall anomalies highlighted in Section 4.1, although a cause–effect relationship remains out of reach at this step of the study. However, the strong control of the boundary forcing on the simulated rainfall (not shown) gives support for the perturbation-induced responses. Moreover, this attempt of physical interpretation of the simulated rainfall regime response to the boundary forcing perturbations shows that the WAM consistency is preserved, providing confidence for the use of this experimental protocol for future work. Furthermore, the contrasted responses are consistent with expectations from current knowledge of the deep convection regions response to a warmer atmosphere and/or a warmer SST (see e.g., [67]), yet with local specifities (e.g., the sahelian dipole). A more complete understanding of the mechanisms involved in the rainfall response should include the existence and importance of (i) feedback mechanisms involving rainfall and e.g., cloud cover, soil moisture, water vapour and (ii) surface–boundary layer interactions through e.g., radiative and turbulent heat fluxes.

## 4. Conclusions

### 4.1. Main Results

In this study, a Surrogate Climate Change (SCC) modelling experiment is performed with the Modèle Atmosphérique Régional (MAR) on West Africa with the aim of assessing the simulated

rainfall response to thermodynamical perturbations. The RCM is driven at its domain boundaries by the ERA-interim reanalyses over a period thought to be representative of the West African Monsoon (WAM) natural variability. This study also addresses the "garbage in, garbage out" issue i.e., the impact of biased boundary forcings on the model response. The model ability to simulate the WAM rainfall regime is first evaluated comparing the model outputs with three independent datasets, namely, a satellite, a rain-gauge and a reanalysis product. The MAR is shown to capture the main features of the WAM rainfall regime at time and space scales relevant for our purpose, that is, considering seasonally and regionally averaged rainfall amounts. Among evidenced flaws are an overall dry bias, which is shared by most RCMs (and GCMs) on this region, and a deficiency in capturing particular features of the interannual variability. Still, this evaluation gives confidence on the ability of the MAR to simulate the present climate of West Africa. The surrogate-type climate change experiments carried out after this initial validation step provide the following clues on the sensitivity of the West African climate to a warmer atmosphere on the one hand, to a warmer ocean on the other hand, as well as to a combination of both:

- Contrasted responses to the imposed perturbations, with an overall drying tendency of $-15\%$ with a warmer atmosphere (T10) and an overall wetting of 17% with a warmer SST (T01), yet with regional discrepancies. Particularly interesting regional features of the rainfall response in these two first experiments are the larger sensitivity of the Guinea region and the sahelian dipole. The combined perturbations experiment (T11) displays a more widespread but smaller wetting tendency of 5%,
- A robust signal, with each experiment resulting in a rainfall anomaly of constant sign in distint synoptic conditions, revealing the strong sensitivity of the rainfall regime to thermodynamical perturbations irrespective of the natural climate variability, most prominently over Guinea,
- A physically consistent model behaviour in response to the boundary forcing perturbations, at least for the mechanisms analyzed and in the range of time and space scales considered,
- Rainfall changes that compare in magnitude with the intrinsic model bias, most prominently over Guinea.

There are two main conclusions to be derived from these results. First, there is a real potential of using SCC scenarios to gain some insights into the main drivers of future rainfall regime changes in this region, especially when considering that both the atmosphere and the ocean will be (and are generally already) getting warmer. Secondly, caution and sagacity are key when using RCM outputs forced by GCM that are known to have difficulties in the tropical band. As a matter of fact, errors originating in the GCM forcing fields for a future climate will have a significant impact on the monsoon dynamics, the associated rain production and other variables like near-surface temperature, of critical importance for subsequent impact studies.

### 4.2. Limitations and Perspectives

The SCC experimental protocol has undeniable advantages. First, it allows for global warming impact studies on a specific region without relying on GCM outputs and their associated uncertaintites. Second, the -purposingly simplistic perturbations allow for a process-based understanding of the model response. Yet, our set of experiments is not deprived of limitations, as discussed below.

### 4.2.1. Study Period

The analysis conducted here highlights the strong interplay between changes in moisture, temperature and wind, and questions their relative contribution to the rainfall changes. Inferring a physical understanding of the boundary forcing perturbations influence on the simulated WAM rainfall regime is an issue, given the complexity of the mechanisms involved. A deeper analysis of within-region (e.g., the E-W sahelian dipole) and interannual (e.g., 1983–1984) differences would be of great interest to identify and compare the relevant mechanisms and their interactions. The MSE budget

framework seems adapted to analyze the model response to boundary forcing perturbations of this nature. For instance, Hill et al. [68] show in an AGCM experiment how the covariance between thermodynamical (i.e., MSE content and gradients) and dynamical (i.e., horizontal and vertical wind speed) anomalies contribute to a strong rainfall dipole in the Eastern Sahel in response to a prescribed SST warming. Another feature worth evaluating is the seasonal shift of the rainy season. Seth et al. [69] suggest a mechanism wherein dry season conditions persist longer before sufficient moisture is eventually supplied, yielding a delayed, shorter and more intense rainy season over West Africa. Ultimately, the experiments performed in this study might offer the possibility to assess the impact of large-scale thermodynamical perturbations on other statistics related to precipitation, although particular care should be taken to remain within the range of temporal and spatial scales where the model best behaves. Exploring these various issues would require a longer period of simulation for (i) the signal-to-noise ratio to be increased and (ii) statistical significance assessment to be achieved.

### 4.2.2. Realism of the Perturbations

In this study, the perturbations of the air and sea surface temperatures are vertically and horizontally homogeneous, respectively, which is quite simplistic. However, we argue that a physically based understanding of the model sensitivity requires, at least at the early stage of the research, this kind of perturbations. The model sensitivity has been well established and, to some extent, proved to be physically consistent. Therefore, applying more realistic perturbations is a next step. For instance, one can think of an altitude-dependent temperature increase: since the temperature profile in the tropics is moist adiabatic -and provided it remains so in a warmer world-, warming causes stability to increase [70]. A step toward more realism and complexity is the Pseudo-Global-Warming Downscaling (PGWDS) framework, consisting in perturbing a reanalyses-driven RCM with the climate change "anomaly" from a GCM run under present and future scenario (i.e., considering only the difference between future and present conditions). Several variables can be considered (e.g., wind, air temperature, SST, geopotential height) in a self-consistent manner (see e.g., [71]). The MAR sensitivity evidenced here suggests promising outcomes from a PGWDS experiment, although it raises questions such as the choice of the GCM signal to be used.

### 4.2.3. Results Model-Dependence

This study has been performed with one model used in one single configuration, making our results highly model-dependent. In West Africa the bulk of rainfall has a convective nature, making the simulated rainfall regime very sensitive to the convective scheme. Besides the representation of precipitation, convection also governs thermodynamics-dynamics interactions. For instance, Hill et al. [68] show the major influence of the convective scheme on an AGCM rainfall response to warm SST anomalies. Critically at stake in this study is the efficiency of convection in communicating aloft the oceanic boundary layer warming and moistening, which strongly depends on the convection parameterization: single- or multi-plumes clouds, entrainment rate calculation, precipitation efficiency and cloud-base closure are among the evidenced sources of differences between schemes. The sensitivity of the MAR response to various convective schemes has been the matter of a study over Belgium [72]: of the evaluated schemes none was shown to perform better, especially for summer precipitation. An analysis of the MAR sensitivity to relevant convective scheme parameters and/or different convective schemes might allow addressing the question of the expected behaviour of the model with a warmer air/SST, at least in terms of rainfall. More generally, the importance of model tuning (e.g., soil moisture recycling efficiency) remains as a critical issue to be addressed for our conclusions to be confirmed. Furthermore, the same experiments performed with other RCM would be of great interest, given the large inter-model spread over this region [73].

4.2.4. Non-Stationary Sensitivity

This study is an attempt at assessing the model boundary forcing bias sensitivity, i.e., the impact of errors in the driving data on the simulated rainfall regime. Although qualitative information on the model predictive skill can be gained from our idealized experiments, the potential change of this sensitivity in a transient climate is not addressed. Therefore, conclusions about the future rainfall evolution have to be considered with care, for at least two reasons:

- precipitation is an "end-of-the-chain" variable: its estimation relies on the faithful representation of many other variables,
- the bulk of rainfall in West Africa is convective, a threshold process.

Hence, it seems of crucial importance to understand the extent to which the boundary forcing bias sensitivity of a regional climate model may change in a different climate. In this respect, Maraun [74] uses a pseudo-reality experimental framework wherein a single GCM–RCM combination is used to identify bias changes in other RCMs from a reference to a future scenario period. Focusing on relevant variables, several mechanisms responsible for the bias changes are identified. Here, it has been shown that the model sensitivity has a clear thermodynamical footprint, especially over Guinea, whereas the respective share of thermodynamical and dynamical contributions to the rainfall response is more difficult to decipher over the Sahel. A better understanding of the respective role of each of these terms in the model sensitivity would allow for a more critical view on a downscalling exercise with a future emissions scenario. Finally, the potential non-stationarity of the boundary forcing bias is itself questionable. Authors in Krinner and Flanner [75] have demonstrated the stationarity of current-generation GCM biases under distinct climate change scenario. They suggest that in-run bias correction of AGCMs (at proper temporal and spatial scales) has the potential for improved RCM driving, a promising way toward a physically based, end-user-oriented understanding of the WAM future evolution.

**Author Contributions:** Conceptualization, H.G., G.C.; methodology, H.G., G.C.; formal analysis, G.C.; investigation, G.C.; resources, H.G.; validation, H.G.; writing—original draft preparation, G.C.; writing—review and editing, H.G., T.L., G.P., T.V.; supervision, H.G., T.L., G.P., T.V.; funding acquisition, T.V. All authors have read and agreed to the published version of the manuscript.

**Funding:** This work was performed using HPC resources from GENCI-IDRIS (Grant 2019-A0060101523). Computations presented in this paper were also performed using the GRICAD infrastructure (https://gricad.univ-grenoble-alpes.fr), which is partly supported by the Equip@Meso project (reference ANR-10-EQPX-29-01) of the programme Investissements d'Avenir supervised by the Agence Nationale pour la Recherche.

**Acknowledgments:** The authors are grateful to the two anonymous reviewers for contributing to improve this paper, in terms of both content and form. The authors also thank Mondher Chekki and Guillaume Quantin for their assistance during the processing of the model and observational data.

**Conflicts of Interest:** The authors declare no conflict of interest.

## Abbreviations

The following abbreviations are used in this manuscript:

| | |
|---|---|
| AGCM | Atmospheric General Circulation Model |
| AOGCM | Atmosphere-Ocean General Circulation Model |
| CMIP | Coupled Model Intercomparison Project |
| ECMWF | European Centre for Medium-Range Weather Forecasts |
| GCM | General Circulation Model |
| GHG | Green-House Gases |
| ITCZ | Inter-Tropical Convergence Zone |
| MAR | Modèle Atmosphérique Régional |
| MSE | Moist Static Energy |
| MSS | Moist Static Stability |

RCM　　　Regional Climate Model
SCC　　　Surrogate Climate Change
SST　　　Sea Surface Temperature
SVAT　　Surface Vegetation Atmosphere Transfer
WA　　　West Africa
WAM　　West African Monsoon

## Appendix A. JAS Rainfall Anomalies Wrt the Control Simulation (T00): Yearly Values

**Table A1.** JAS cumulative rainfall (mm) for the control simulation (T00) and sensitivity experiment anomalies with respect to the control simulation. Relative values (%) are displayed in parenthesis.

| Years | Anomalies | Guinea | Sahel | W Sahel | E Sahel |
|---|---|---|---|---|---|
| 1983 | T00 | 340.6 | 303.2 | 280.1 | 327.7 |
| | T10-T00 | −79.7 (−23.4) | −21.5 (−7.1) | −45.6 (−16.3) | 4.15 (1.3) |
| | T01-T00 | 104.2 (30.6) | 14.3 (4.7) | 29.9 (10.7) | −2.2 (−0.7) |
| | T11-T00 | 34.2 (10.1) | 6.5 (2.15) | 3.6 (1.3) | 9.6 (−2.9) |
| 1984 | T00 | 360.4 | 224.3 | 184.4 | 266.6 |
| | T10-T00 | −69.8 (−19.4) | −8.1 (−3.5) | −20.9 (−11.3) | 5.5 (2.1) |
| | T01-T00 | 99.9 (27.7) | 2.0 (0.9) | 17.8 (9.7) | −14.8 (−5.5) |
| | T11-T00 | 32.9 (9.12) | 9.3 (4.1) | 9.8 (5.3) | 8.7 (3.3) |
| 1993 | T00 | 324.0 | 272.8 | 255.3 | 291.3 |
| | T10-T00 | −72.5 (−22.4) | −19.1 (−6.9) | −33.4 (−13.3) | −3.3 (−1.1) |
| | T01-T00 | 100.5 (31.0) | 24.6 (9.0) | 40.8 (16.0) | 7.3 (2.5) |
| | T11-T00 | 37.9 (11.7) | 23.8 (8.7) | 28.4 (11.1) | 18.9 −6.5) |
| 1994 | T00 | 297.4 | 269.3 | 206.8 | 335.5 |
| | T10-T00 | −70.6 (−23.8) | −32.0 (−11.9) | −51.6 (−24.9) | −11.2 (−3.2) |
| | T01-T00 | 113.4 (38.1) | 12.0 (4.4) | 25.6 (12.4) | −2.5 (−0.8) |
| | T11-T00 | 38.6 (13.0) | 16.3 (6.1) | 9.61 (4.64) | 23.4 (7.0) |
| 2010 | T00 | 340.4 | 330.7 | 305.6 | 357.4 |
| | T10-T00 | −69.2 (−20.3) | −12.6 (−3.8) | −30.9 (−10.1) | 6.8 (1.9) |
| | T01-T00 | 104.3 (30.7) | 7.25 (2.2) | 17.5 (5.7) | −3.6 (−1.0) |
| | T11-T00 | 39.0 (11.5) | 18.2 (5.5) | 28.0 (9.2) | 7.8 (2.2) |
| 2011 | T00 | 327.5 | 287.1 | 255.8 | 320.4 |
| | T10-T00 | −89.2 (−27.2) | −30.4 (−10.6) | −44.5 (−17.3) | −15.4 (−4.8) |
| | T01-T00 | 92.1 (28.1) | 1.9 (0.7) | 24.2 (9.4) | −21.7 (−6.8) |
| | T11-T00 | 15.6 (4.9) | 15.0 (5.2) | 23.7 (9.3) | 5.76 (1.8) |

## Appendix B. JAS Total Rainfall Comparison for the T01 experiment

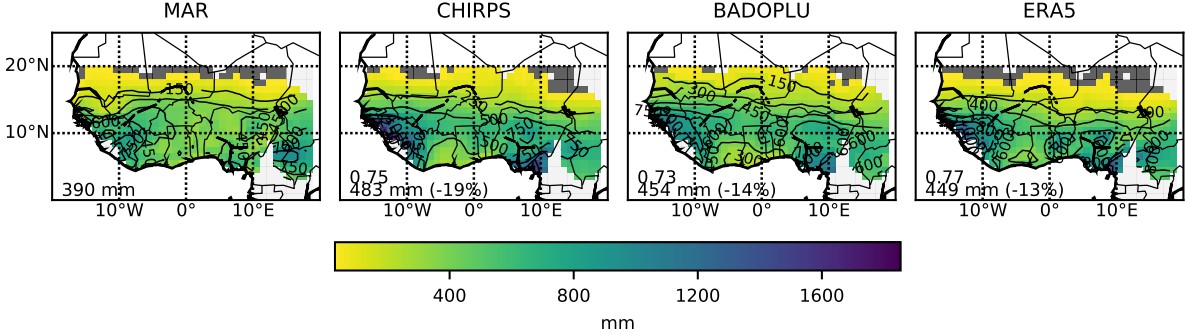

**Figure A1.** As in Figure 2 but for the T01 sensitivity experiment.

## Appendix C. 925 hPA Temperature Sensitivity Experiment Anomalies

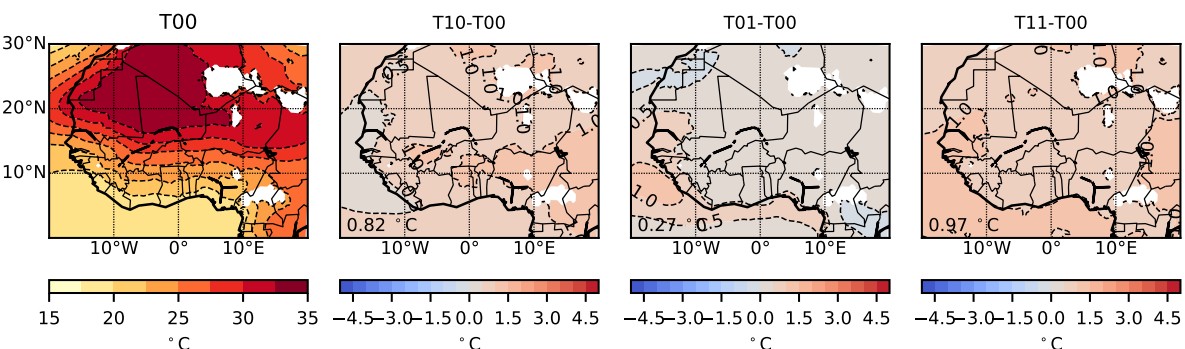

**Figure A2.** JAS 925 hPa temperature for the control simulation and sensitivity experiment anomalies.

## Appendix D. 925 hPA Relative Humidity Sensitivity Experiment Anomalies

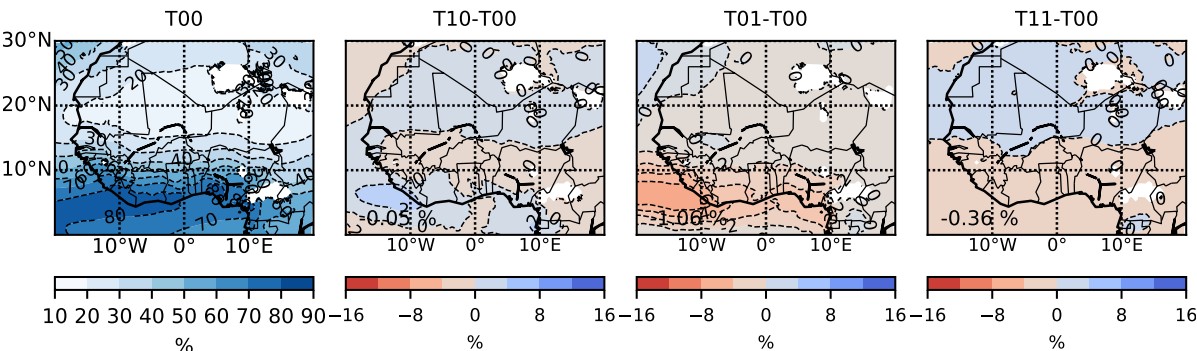

**Figure A3.** JAS 925 hPa relative humidity for the control simulation and sensitivity experiment anomalies.

## Appendix E. MSE Terms Profiles

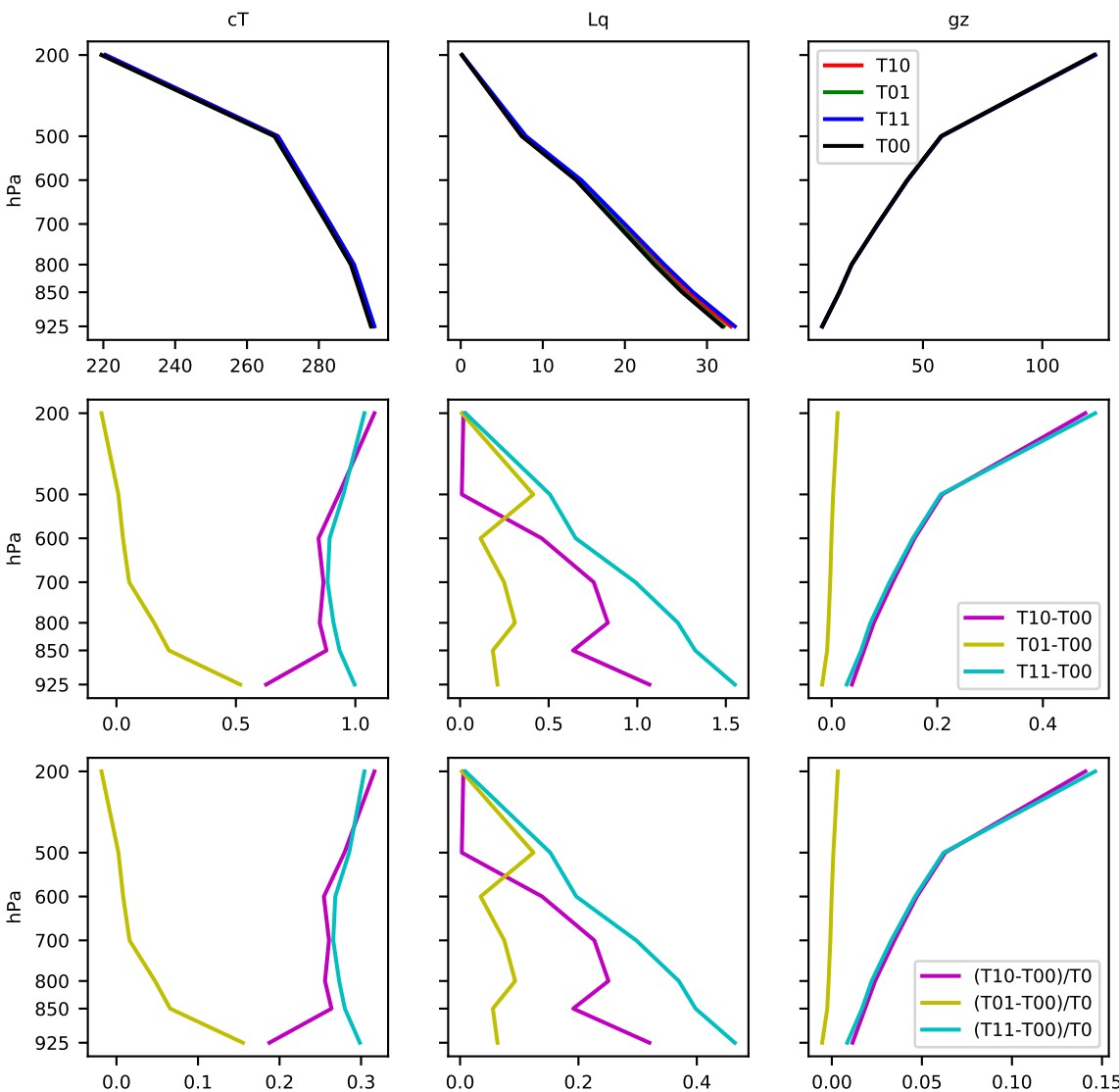

**Figure A4.** Guinea regional mean JAS MSE terms for the control run and sensitivity experiment (J g$^{-1}$, **top**), anomalies (J g$^{-1}$, **center**) and changes relative to total MSE control value (%, **bottom**). All values are study period averages.

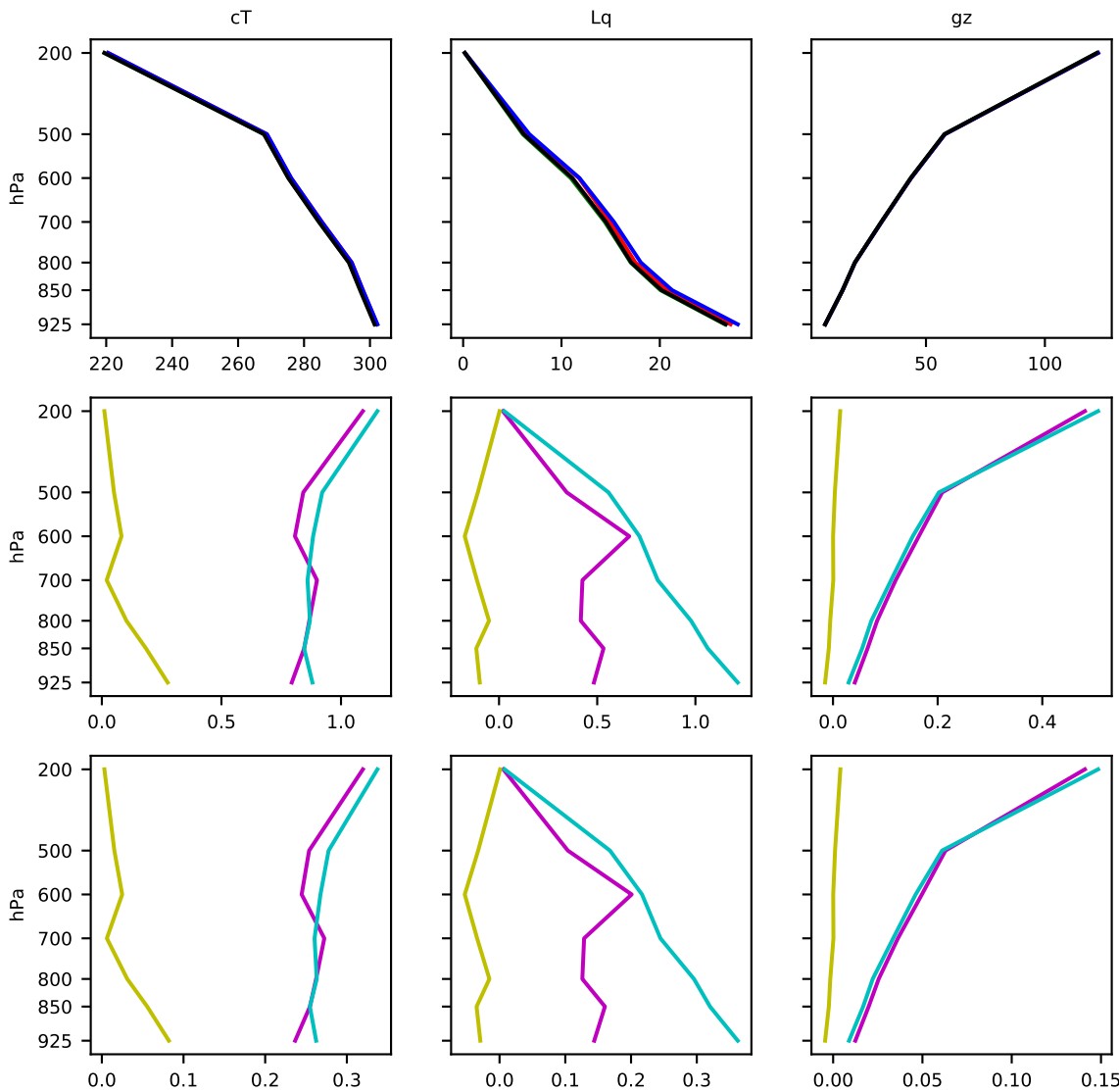

**Figure A5.** Same as Figure A4 for the Sahel. Note the different horizontal axis values.

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
