# Peer review of "A Boundary Forcing Sensitivity Analysis of the West African Monsoon Simulated by the Modèle Atmosphérique Régional"

_atmosphere, doi:10.3390/atmos11020191_

Round 1

Reviewer 1 Report

The authors study the response of a regional model to its boundary forcing in the context of an increase of temperature due to climate change. The studied region: West of Africa, is of high interest for many reasons.

The study is sound and the manuscript is clear. We recommend to publish the paper in the present form, except for a few sentences or expressions that could be rephrased for better clarity.

Line 106: "grading", "ranking" will be more appropriate

Line 108: Various approaches have been proposed to that end, usually making use of reanalyses as boundary forcing fields, assumed to minimize the biases at the boundaries of the simulation domain. This phrase is not grammatically correct.

Line 112: "run under a climate scenario"

Line 114: They conclude that while boundary forcing and inter-model errors are in the same order – these errors depending on the season considered is not clear. Please rephrase.

Line 120: "by isolating it" what does "it" refer to? Please rephrase.

Line 128: "linked to"

Line 138: The MAR acronym is used but only defined at line 144

Line 141: "has proven instructive to studies"

Line 153: "mindful use" informed might be more appropriate

line 165: work of Betchold

line 265: "The later"

Line 274: "with values for each data set" does this mean that only rainy pixels are included in the verification?

Table 1: The "T00" acronym appears for the first time here. It hasn't been defined. It should be included in the definitions at lines 227.

Line 312: The table number is missing

Line 325: "in parenthesis"

Line 351: The table number is missing

line 386: linked to a dynamical mechanism

line 393: the increased moisture holding capacity of the atmosphere (T10) only results

line 557: "Results model-dependence" is incomplete, please expand.

Author Response

We are grateful to the reviewer for his analysis and for providing us with useful hints on how to improve our article. Please find attached a point by point reply to the comments.

Reviewer 2 Report

Overview

This paper uses a regional climate model (RCM), driven by reanalysis data, to conduct a number of sensitivity experiments focusing on the West African monsoon.  The work involves the Surrogate Climate Change (SCC) methodology, whereby the boundary forcing conditions (such as air temperature, SST and a combination of both) are changed, and the response is assessed. 

Major comments

Nice paper.  Clearly and well written, with a good introduction providing detailed background material and motivation.  The paper then presents the model and the methodology, before evaluating the model output against observations.  The results are then presented, before the paper summarises/concludes.  The methodology is sound and robust, and the results are nicely summarised by the conclusions and visualised by the (refreshingly) clear figures.  The work is novel because although other studies exist conducting sensitivity experiments over this region, this is the 1st to use the SSC methodology in this context.  I therefore recommend publication, with the following suggestions.

In the evaluation section, the authors only compare the simulated data with 2 observational datasets. The paper would benefit from the inclusion of other datasets, of which there are many over Africa, either in-situ measurements or satellite-derived rainfall estimates (e.g. ARC2, RFE, TAMSAT).  Alternatively, global datasets e.g. TRMM, GPCP could and indeed should be included.  This would give a much better idea of how well the model performs under present-day conditions. In the experiment results section, the physical mechanisms (i.e. the dynamics) are only really mentioned in a couple of paragraphs at the end, almost as an afterthought. I would like to see this section extended, to include other fields e.g. winds at various levels, convergence/divergence, pressure etc, to give a much clearer idea of how the West African monsoon response to the various perturbations is occurring.

Minor comments

Abstract: a sentence or two of motivation is needed at the beginning, to say why this is a “crucial societal and scientific challenge”. Introduction: MAR needs to be defined further up (line 138) Introduction: line about MAR should be moved down to methodology section (lines 144-149) Results: see above for comment about observational data Results: figure 11 needs to be bigger Overall: careful proofreading is needed, because the paper is littered with minor grammatical errors. None of these get in the way of comprehension, and are perfectly excusable as English is probably not the authors’ first language, however should be corrected before publication.  Careful proofreading is also needed because in several places, table and figure numbers are missing such as ”Table ??” (e.g. line 313)

Author Response

(The authors gave the same response as above.)
